# Multilayer Model in Soil Moisture Content Retrieval Using GNSS Interferometric Reflectometry

**DOI:** 10.3390/s23041949

**Published:** 2023-02-09

**Authors:** Jie Li, Xuebao Hong, Feng Wang, Lei Yang, Dongkai Yang

**Affiliations:** 1Electronic Information Engineering, Beihang University, Beijing 100191, China; 2School of Information Science and Technology, University of Jinan, Jinan 250022, China

**Keywords:** global navigation satellite system–interferometric reflectometry, soil moisture content, multilayer, Hilbert transform, sink model, reverse osmosis model

## Abstract

The global navigation satellite system–interferometric reflectometry (GNSS-IR) was developed more than a decade ago to monitor soil moisture content (SMC); a system that is essentially finished has emerged. The standard GNSS-IR model typically considers soil to be a single layer of medium and measures the average SMC between 1 and 10 cm below the soil surface. The majority of the SMC is not distributed uniformly along the longitudinal axis. This study is based on a simulation platform and suggests a SMC-stratified measurement model that can be used to recover the SMC at different depths in the sink and reverse osmosis to address the issue that conventional techniques cannot accurately measure soil moisture at different depths. The soil moisture of each layer was assessed by utilizing the GNSS signals reflected by various soil layers, and this study employed total transmission when the vertical linearly polarized component of the electromagnetic wave was conveyed by the GNSS signal reflected by the soil. This work employed the Hilbert transform to obtain the interference signal envelope, which increases the visibility of the interference signal’s “notch” and reduces the burr impact of the interference signal brought on by ambient noise. The accuracy of the SMC measurement at the bottom declines due to the soil’s attenuation of the GNSS signal power, but the correlation between the predetermined value and SMC retrieved by the GNSS-IR multilayer SMC measurement model similarly approached 0.92.

## 1. Introduction

Soil moisture content (SMC) is used to describe how wet the soil is. SMC is an important factor to consider while researching the water cycle and terrestrial energy. It can alter the conversion of atmospheric precipitation, such as rain and snow, into evaporation and infiltration, as well as the proportion of the surface runoff, in addition to changing the ratio of net radiant energy to latent heat and sensible heat [1]. The global navigation satellite system–interferometric reflectometry (GNSS-IR) is a new remote sensing technology that uses the reflected signals of navigation satellites on land and ocean surfaces to invert physical parameters, such as SMC [2], vegetation growth status [3], sea level [4], and snow depth [5,6]. Using GNSS-IR technology to measure soil moisture can reduce costs, increase the measurement area, and reduce damage to land and crops. Compared with the traditional probe method and other methods, the GNSS-IR technology equipment is simple to build and has a long service life, which makes the measured data more conducive to the application of water cycle monitoring, agriculture, and forestry [7].

The idea to monitor soil moisture with GNSS interferometric signals was initially put forward by Larson et al. in 2008. Since then, researchers from all around the world have studied this technique [8]. The method for determining the signal-to-noise ratio (SNR) of the horizontal and vertical polarization signals was proposed in 2014 by altering the polarization mode of the receiving antenna [9]. Some researchers have looked at the retrieval of soil moisture under vegetation cover to lessen the impact of vegetation on soil moisture retrieval [10,11,12,13]. The spectral estimate approach inverts the real vertical height between the GNSS antenna and the reflector. The plant canopy is believed to have taken the position of the soil surface as the principal reflector if there is a considerable difference between the height and the height of the installation. As a result, the data from this time period may be discarded [11]. According to [14], the multipath peak frequency’s sensitivity to field soil water was determined. The multipath peak frequency may be utilized to accurately assess the signal’s spectrum because it depends on the field’s soil wetness. The varying attenuation of the GNSS signal propagating under varied soil moisture may also be used to perform the soil moisture inversion. The results indicate that the greatest depth at which the GNSS signal may penetrate soil moisture is around 21 cm and that the relationship between the signal’s penetration depth and soil moisture is roughly linear [15]. To retrieve SMC, Hong et al. recommended employing GNSS pseudo-interference reflectometry (GNSS-PIR) [16]. They established that the GNSS-PIR measurement is more accurate than the conventional GNSS-IR measurement by combining the signal received by the right circular polarization (RHCP) and left circular polarization (LHCP) antennas to generate an oscillatory carrier noise ratio (CNR) waveform. An approach for aberrant phase identification and correction using a multi-satellite combined SMC retrieval system was suggested in light of the phase anomaly induced by the low-quality SNR.The SNR data were split into trend and modulation components using a wavelet transform, and all satellite aberrant phases were then found and corrected. The outcomes demonstrated that their algorithm can efficiently identify and correct abnormal phases [17].

In addition to RHCP and LHCP, GNSS antennas can also utilize linearly polarized antennas to catch reflected signals. For instance, cell phone GNSS navigation antennas are often linearly polarized, according to [18]. The test demonstrated that the Xiaomi Mi 8 smartphone may pick up the GNSS signal reflected by the ground and interfere with the direct transmission. The obtained antenna RMSE height is 1.9 m, and the results demonstrate that the GNSS interference signal captured by linear polarization may persist for a longer satellite elevation angle range [19]. Two smartphones were connected to the upper and lower sides of the plane, allowing for the isolation and recording of the direct and reflected GNSS carrier noise density ratios as well as the remote sensing of land surface data. The results revealed that the reflected signals exhibited geographical correlations in line with surface properties, including the pond’s high reflectivity, susceptibility to different types of land crops, and soil wetness [20]. When an electromagnetic wave shines on a dielectric material—that is, when it passes through the upper dielectric and is reflected by the lower dielectric—it exhibits the phenomena of the complete transmission. Moreover, this phenomenon has something to do with how electromagnetic waves are polarized. The GNSS signal’s vertical polarization will result in the complete transmission phenomena. The inaccuracy was less than 8% when Rodriguez et al. utilized this characteristic to invert soil moisture under foliage cover [21].

The theoretical approaches and simulation outcomes of SMC multi-layer measurements in the sink and reverse osmosis modes for the asymmetric water distribution in the soil are presented in this study to retrieve SMC at various depths. The complete transmission of the electromagnetic wave incident on the medium’s surface—vertically linearly polarized—was used to capture the GNSS signals reflected by soil moisture in various layers. First, a noise-free simulation of the soil multilayer interference signal was performed, and Lomb–Scargle periodogram (LSP) calculations were used to determine the spectrum features of the reflected signal. The results demonstrate that the soil moisture beneath can be reflected by the distinctive features of the GNSS interference signal “notch” with vertical linear polarization. When compared to the reverse osmosis model, the sink model’s reflectivity is more significantly impacted by the soil moisture’s interval change than the reverse osmosis model. In addition, the mode layer number essentially has no impact on the power reflectivity, regardless of the setting. Finally, we examine the noise-layered model’s properties. In this investigation, the envelope of the reflected signal is extracted using the Hilbert transform. The final spectrum of the reflected signal agrees with the noise-free modeling results. This paper is based on simulation, and is organized as follows: the Section 2 presents the theoretical model, the Section 3 presents the simulation study, and the Section 4 and Section 5 describe the results and conclusions.

## 2. Theory

Both the direct signal from line-of-sight propagation and the signal reflected off the reflecting surface may be picked up by the geodetic GNSS receiver. The reflecting surface affects these reflected GNSS signal qualities, and effective signal extraction and processing may be utilized to depict the physical characteristics of the reflective surface. When two electromagnetic waves of the same frequency and phase arrive at the same location in space, they will interfere with one another. In ground-based GNSS, there is little change in frequency between direct and reflected signals. An SNR data model for the interference of a single reflected signal and a direct signal is shown in the literature [8]. The direct signal is separately interfered with by the signal reflected by each layer of soil, and the SNR data model of multilayer soil interference signal is generated as Equation (Equation 1) shows.
(1)$$SNR=A_{d}^{2}+\sum_{i=1}^{n}A_{m_{i}}^{2}+2\sum_{i=1}^{n}A_{d}A_{m_{i}}\cos\varphi_{i}$$
where $A_{d}$ and $A_{m_{i}}$ represent the amplitude of the direct and reflected GNSS signals from the *i*th layer. Moreover, $\varphi_{i}$ represents the phase difference between the direct and reflected signals from the *i*th layer.

The signal reflected by the multi-layer soil is picked up by the GNSS receiver, which interferes with the directed signal; the oscillations naturally occur as a result. The schematic representation of the GNSS signal’s propagation route via the multi-layer soil model is shown in Figure 1. Figure 1 demonstrates that when a GNSS signal interacts with the dielectric surface, both penetration and reflection take place. The signal will continue to transmit after being reflected.

In general, there are two types of soil surface moisture distributions: sinking types (such as rainfall), in which the humidity falls with depth; and reverse osmosis types (such as river tidal flats), in which the humidity increases with depth.

Satellite electromagnetic waves may be thought of as plane waves for a reflection route. The travel delays of the direct and reflected signals are equal to the differences between the red and green lines and may be expressed according to [22], based on the geometric connection in Figure 1:(2)$$\delta_{i}=\frac{H_{i}}{\sin\theta }-\frac{H_{i}}{\sin\theta }\cos2\theta_{i}$$
where *H* represents the vertical height of the GNSS receiver antenna from the soil surface, and θ represents the elevation angle of the satellite. After simplifying Equation (Equation 2), we can have:(3)$$\delta_{i}=2H_{i}\sin\theta_{i}$$
the oscillation frequency of the interference signal SNR with the sine of the elevation angle as the horizontal axis can be calculated by the following formula [22]
(4)$$f_{m_{i}}=\frac{d(\delta_{i}/\lambda )}{d\sin\theta_{i}}=\frac{2H_{i}}{\lambda }$$
where λ represents the wavelength of the electromagnetic wave carrying the GNSS signal.

Equation (Equation 4) shows that the vertical height of the GNSS antenna above the soil surface affects the oscillation frequency of the interference signal. For generally level ground, its roughness has minimal impact on the signal frequency, and the primary interference comes from the nearby flora.

Refraction and reflection occur when an electromagnetic wave illuminates the boundary between two media types. The following are the definitions of the vertical polarization’s Fresnel reflection and refraction coefficients:(5)$$\Gamma_{⊥}=\frac{\eta_{2}\cos\theta_{d}-\eta_{1}\cos\theta_{\tau }}{\eta_{2}\cos\theta_{d}+\eta_{1}\cos\theta_{\tau }}$$
and
(6)$$T_{⊥}=\frac{2\eta_{2}\cos\theta_{d}}{\eta_{2}\cos\theta_{d}+\eta_{1}\cos\theta_{\tau }}$$
where $\eta_{1}=\sqrt{\mu_{1}/\varepsilon_{1}}$ and $\eta_{2}=\sqrt{\mu_{2}/\varepsilon_{2}}$ represent the electromagnetic wave impedance coefficient, respectively. Moreover, $\mu_{1}$ and $\varepsilon_{1}$ are the permeability and permittivity of the air above the soil, respectively. $\mu_{2}$ and $\varepsilon_{2}$ are the permeability and permittivity of the soil, respectively. $\theta_{d}$ and $\theta_{\tau }$ represent the incidence angle and the refraction angle, respectively.

Furthermore, the Fresnel reflection and refraction coefficients of horizontal polarization can be defined as:(7)$$\Gamma_{\|}=\frac{\eta_{1}\cos\theta_{d}-\eta_{2}\cos\theta_{\tau }}{\eta_{1}\cos\theta_{d}+\eta_{2}\cos\theta_{\tau }}$$
and
(8)$$T_{\|}=\frac{2\eta_{2}\cos\theta_{d}}{\eta_{1}\cos\theta_{d}+\eta_{2}\cos\theta_{\tau }}$$

The reflection angle, refraction angle, and incident angle satisfy the relationship:(9)$$\left\{\begin{array}{c} \theta_{r}=\theta_{d}\\ \frac{\theta_{\tau }}{\theta_{d}}=\frac{\mu_{1}\varepsilon_{1}}{\mu_{2}\varepsilon_{2}} \end{array}\right.$$

The electromagnetic wave propagation in the soil will cause signal power attenuation. We set the GNSS signal power in an open space as $p_{a}$; the power after penetrating the soil can be calculated as:(10)$$p_{s}=p_{a}e^{-\alpha l_{s}}$$
where $l_{s}=d_{s}/\cos\theta_{i}$ and α can be calculated as:(11)$$\alpha =\epsilon_{s,imag}\epsilon_{0}2\pi f\sqrt{\frac{\mu_{0}}{\epsilon_{s,real}\epsilon_{0}}}$$

The phenomenon of the entire transmission may occur when electromagnetic waves are irradiated into the interfaces of two distinct media types, as can be observed from the literature cited in [3]. The critical angles of total transmission for the vertical and horizontal polarizations differ, and the incidence angle when the two polarizations are transmitted may be computed as follows:(12)$$\sin\alpha_{v}=\sqrt{\frac{\mu_{1}\varepsilon_{2}-\mu_{2}\varepsilon_{1}}{\varepsilon_{1}\left(\mu_{1}^{2}-\mu_{2}^{2}\right)}\mu_{2}}$$
and
(13)$$\sin\alpha_{h}=\sqrt{\frac{\mu_{2}\varepsilon_{1}-\mu_{1}\varepsilon_{2}}{\mu_{1}\left(\varepsilon_{1}^{2}-\varepsilon_{2}^{2}\right)}\varepsilon_{2}}$$
where α represents the critical incident angle and is complementary to the satellite elevation angle. They mean the permeability and permittivity of the upper and lower soil when we discuss the signal propagating inside the soil.

We substitute the air and soil permittivity and permeability when the soil humidity is from 5% to 40%, $\alpha_{v}$, generally between 65° and 85°, and $\alpha_{h}$ close to 0°. The receiver can only now receive direct signals from line-of-sight propagation due to the entire transmission phenomena of vertical polarization at high incidence angles (i.e., low satellite elevation angles). In order to determine the soil moisture layers, the GNSS signals reflected by distinct soil layers can be independently collected using the vertical linear polarization total transmission characteristic.

The interference signal’s SNR oscillation waveform, which was captured using a vertically linearly polarized GNSS antenna, is shown in Figure 2. The simulation is set to sink mode, and a 20 cm layer of soil moisture is lowered from 30% to 20%. In Figure 2, there is a “notch” where the satellite elevation angle is 15°, with its amplitude significantly decreasing in comparison to other height angles. The larger portion of Figure 2 demonstrates the presence of three distinct frequency components relating to the satellite’s elevation angle at the problem location. The expanded portion of the picture demonstrates that there are three distinct frequency components related to the satellite’s elevation angle at the defect site.

We divide the upper and lower sides of Equation (Equation 12) by $\varepsilon_{1}$, and it can be rewritten as:(14)$$\sin\alpha_{v}=\sqrt{\frac{\mu_{1}k-\mu_{2}}{\left(\mu_{1}^{2}-\mu_{2}^{2}\right)}\mu_{2}}$$
where $k=\varepsilon_{2}/\varepsilon_{1}$. As the soil moisture drops, the permittivity for the sink mode falls, and the ratio *k* rises. The angle at which full transmission takes place will decrease with the drop in $\sin\alpha_{v}$, according to the sine function’s monotonicity between 0° and 90°. The interference signal SNR created by the reflected signal and the direct signals at the bottom, middle, and surface soil, respectively, are shown in Figure 2’s magnified Section 1, Section 2 and Section 3.

The outcomes of the comparison between the single-layer and multi-layer models are shown in Figure 3. In contrast to the multi-layer model, the single-layer model only has one “notch” in the SNR of the interference signal. With the aid of the vertical polarization’s total transmission characteristic, it is possible to measure the soil moisture with a depth gradient.

## 3. Simulation Analysis

### 3.1. Unchanging Number of Layers

When monitoring soil moisture using GNSS-IR, it is first necessary to separate the direct and reflected GNSS signals. Given that the amplitude of the reflected signal is substantially smaller than that of the direct signal ($A_{d}\gg A_{m}$), the interference signal’s low-amplitude oscillation is brought on by the reflected signal. This allows the reflected component to be acquired by subtracting the direct component from the interference signal, and the direct signal component to be produced via low-order binomial fitting.

After decoupling the direct reflected signal, Figure 4 displays the SNR waveform of the reflected signal. The oscillation frequency is mostly determined by the reflected signal, and the defect position is the same as that of the original interference signal, according to the simulation results of the sink model shown in the figure.

In actuality, the two models of sinking and reverse osmosis are the only ones that can adequately describe how soil surface moisture fluctuates from top to bottom in various regions. Soil moisture changes are assumed to be linear in this study’s simulation. In this study, the reflectivity (amplitude of directed and reflected signal) results of soil moisture from 20% to 30% and 80% are provided for two alternative water distribution models. Other simulation parameters show that the number of layers is set to 3, the antenna height is assumed to be 5 m, the satellite elevation angle is changed to 5° sim 80°, the GNSS antenna is used, the mean square height of soil surface fluctuation is set to 5 cm, and the GPS L1 frequency point signal is used.

Figure 5 displays the reflectivity results for the two modalities; the right panel displays the results from reverse osmosis, while the left panel displays the results from the sink model. When the satellite elevation angle is high, the top layer of soil reflects the majority of the GNSS signal, according to the sink model. The interference signal presents the “notch” position between 10° and 20° of the satellite elevation angle as the enlarged part of Figure 2 shown. The left panel demonstrates that the peak of reflectance also increases in size the lower the satellite elevation angle is when the full transition occurs and the soil surface moisture is higher. Additionally, the rise in the peak of power reflectivity steadily diminishes as the surface soil moisture increases. We simply modified the bottom soil moisture when modeling the reverse osmosis model, leaving the soil top moisture untouched. The right panel, for instance, demonstrates that altering the soil’s moisture content has no impact on the trend of reflectance with satellite elevation angle. This is due to the fact that the moisture of the topsoil has a greater impact on the signal attenuation and overall intensity of the reflected signal than the moisture of the bottom soil.

Multiple reflective surfaces exist in soil with unequal water distribution, creating several reflecting signals. The peak value and peak position of the LSP spectrum of the reflected signal are variable as a result of the varying distance between the reflecting surface and the soil surface. The results of the LSP spectrum estimate for various SMC variation ranges under two distinct modes are shown in Figure 6. Figure 6’s SMC distributions all exhibit multi-peak properties. The maximum peak power rose with the rise in surface SMC when the soil was in sink mode, and the moisture on the soil surface was higher than that at the bottom. The maximum peak power rose with the rise in surface SMC when the soil was in sink mode, and the moisture on the soil surface was higher than that at the bottom. The moisture on the soil top was lower when the soil was in the reverse osmosis mode than it was at the bottom, and the position of the greatest peak power relative to the measured height was also diminished. Additionally, the fraction of electromagnetic wave transmission was decreased, and the power to reach the second layer was decreased, according to Equation (Equation 6). As a result, the peak power was drastically decreased.

We set the wave impedance of the air and the first and second layers of soil as $\eta_{a}$, $\eta_{s1}$, and $\eta_{s2}$, respectively. Then the reflected signal power of the second layer of soil can be written as:(15)$$p_{s2}=p_{a}\frac{2\eta_{s1}}{\eta_{s1}+\eta_{a}}\frac{\eta_{s2}-\eta_{s1}}{\eta_{s1}+\eta_{s2}}\frac{2\eta_{a}}{\eta_{s1}+\eta_{a}}e^{-2\alpha l_{ls}}$$

Figure 7 displays the outcomes of a theoretical Equation (Equation 15) computation. The sink model is in the panel on the left. In the top layer of soil, we change the SMC. Reverse osmosis is modeled in the right panel, and the bottom SMC is changed. Three layers total were created through simulation in both models.

According to the sink model, when SMC in the first layer rises, the strength of the signal transmitted into the second layer falls and the reflectivity of the second layer also decreases. In the reverse osmosis model, when the bottom SMC rises, the second layer of the soil’s SMC rises as well, increasing reflectivity. Furthermore, the second layer of the soil’s reflectivity tends to stabilize as the bottom SMC rises.

Because the energy of the electromagnetic wave is already low when it penetrates the soil to a depth of 21 cm, the penetration depth used in remote sensing is 10.5 cm [15]; therefore, the increase in the underlying soil moisture has no impact on the power reflectivity for the reverse osmosis mode.

### 3.2. Variable Layer

This study adjusts the number of layers in the model for the simulation analysis, and the results are shown in Figure 8, in order to analyze the impact of the number of layers in different models on the outcomes. The graphic demonstrates that the reflectance of the two modes is mostly independent of the number of layers. As a result, the number of layers is no longer studied in this study.

### 3.3. Added Noise

The interference signal SNR recorded in the experiment will really be impacted by the system and environmental noises. We introduced noise into the measured SNR to bring it closer to the actual experiment. A mix of white noise and flicker noise seems to be the best model for the noise characteristics of the GPS, according to the [23,24]. The black line in Figure 9 depicts the SNR of the interference signal after adding a group of Gaussian white noise with an average value of 1 dB.

According to Figure 9, there are several burrs in the time series of the interference signal SNR after the addition of Gaussian white noise, which will impair the measurement of the notch of the vertical signal transmission. In this work, the interference signal’s envelope is obtained using the Hilbert transform to lessen the impact of burr. Hilbert transforms take the unprocessed signal as:(16)$$x\left(t\right)=A\left(t\right)cos\left(2\pi ft+\varphi_{r}\left(t\right)\right)=A\left(t\right)cos\left(wt+\varphi_{r}\left(t\right)\right)$$

Hilbert transform:(17)$$\hat{x}\left(t\right)=H\left[x\left(t\right)\right]=\frac{1}{\pi }\int_{-\infty }^{\infty }\frac{x\left(\tau \right)}{t-\tau }d\tau$$
and its frequency characteristic is:(18)$$\hat{X}\left(jw\right)=X\left(jw\right)H\left(jw\right)$$
where
(19)$$H\left(jw\right)=\left\{\begin{array}{c} -jw\geq 0\\ +jw\leq 0 \end{array}\right.$$

The Hilbert transform, which is identical to an orthogonal filter, shifts the phase by −90° for all positive frequency components and +90° for all negative frequency components, as shown by the equation above, but the amplitude remains constant. As a result, an analytical signal that has the original reflected oscillation signal acting as its real component and the signal’s Hilbert transformation as its imaginary component may be created.
(20)$$\tilde{x}\left(t\right)=x\left(t\right)+j\hat{x}\left(t\right)$$
substitute Equation (Equation 16) into Equation (Equation 20): (21)$$\tilde{x}\left(t\right)=A\left(t\right)\cos\left(wt+\varphi_{r}\left(t\right)\right)+jA\left(t\right)sin\left(wt+\varphi_{r}\left(t\right)\right)=A\left(t\right)e^{j\left(wt+\varphi_{r}\left(t\right)\right)}=A\left(t\right)e^{j\varphi_{r}\left(t\right)}e^{jwt}$$
where $e^{jwt}$ represents the complex carrier signal, and $e^{j\varphi_{r}\left(t\right)}$ represents the complex envelope. Finally, the envelope signal can be obtained by calculating the absolute value of the analytic signal.

The SNR envelope of the interference signal, as determined by the Hilbert transform, is shown as a red line in Figure 9. It is clear that the signal SNR following the Hilbert transformation preserves the frequency and phase properties of the original interference signal while lessening the impacts of roughness on the amplitude. The signal through the Hilbert transform has a stronger resemblance to the SNR sequence of the interference signal without noise than in Figure 4.

The high-frequency components of the interference signal will be present given the presence of noise. In this study, distinct frequency components are extracted and high-frequency components are eliminated using the empirical mode decomposition (EMD) approach. EMD breaks down the signal into a number of intrinsic mode functions based on the time scale of the local aspects of the signal (IMF). The results of the EMD decomposition and its spectral properties are shown in Figure 10. The 4 panels on the right represent the matching spectrum, while the 4 panels on the left represent the IMF1, 3, 6, and 7 components. The spectrum shows that IMF1’s high-frequency component includes a large number of spectral peaks. Low-frequency components include IM3, 6, and 7, of which IMF6 and 7 should be direct current (DC) components due to their lower frequencies. IMF3’s spectrum peak emerges at 5 m which shows that the soil surface is where the GNSS signal for this component is reflected. Additionally, other IMF components did not exhibit spectral peaks at approximately 5 m, and the SNR of the IMF3 exhibits a “notch” when the satellite elevation angle is around 18°.

As shown in Figure 11, spectral graphs of two different modes under different soil moisture changes are drawn in this study. Similar to Figure 6, the spectrum of the two modes and the simulation results without noise show the same trends.

## 4. Results

We set the minimum humidity at 20% cm^3^/cm^3^ and increase it by 1% cm^3^/cm^3^; the maximum humidity is 80% cm^3^/cm^3^, and increases by 1% cm^3^/cm^3^. Each group is simulated 1000 times, and the results are displayed in Figure 12. The reverse osmosis model is on the left, and the sink model is on the right; together they represent the scatter plot of set soil moisture and the results obtained by the model suggested in this study for the two modes. The results of the measurements are somewhat lower than the predetermined soil moisture, according to both models. Additionally, when SMC increased, the relationship between the set SMC and the retrieved results was less correlated. This is due to the fact that when SMC rises, the GNSS signal’s attenuation when passing through soil rises and the accuracy of data processing reduces.

The simulation results in this study demonstrate that although the layers set in the model have little impact on the variation of reflectance with the satellite elevation angle, the variable range of soil moisture from the surface to the bottom has a substantial impact. The entire transmission properties of vertically linearly polarized electromagnetic waves in GNSS-IR technology allow the three-layer soil model to estimate stratified moisture for conventional surface soil.

The two soil moisture-layered models developed for this work generally yield satisfactory retrieval results, with a 0.92 correlation between the measured results and the set SMC. Ground-based experiments were not performed in this study to time the restrictions; however, they will be in subsequent studies.

## 5. Discussion and Conclusions

It has taken decades for GNSS-IR to evolve into a fairly developed approach for retrieving SMC. The goal of the current study is to lessen the impact of vegetation, snow cover, and soil texture. Moreover, the majority of research locations are farms. Moreover, progressively switching from the ground scene to the space scene was the platform.

Another crucial component in the study of meteorology is the study of tidal flat SMC. The purpose of this study is to investigate how soil moisture is distributed unevenly due to groundwater penetration and rainfall. The model developed in this work can accurately quantify SMC at various depths using simulation analysis, and it has some value for researching how SMC changes during rainfall and groundwater infiltration.

In this study, the retrieval of SMC was carried out by using the total transmission of electromagnetic waves with vertical polarization characteristics in GNSS signals on the soil surface. Moreover, a multi-layer model for soil moisture measurements using GNSS-IR was established, including sink and reverse osmosis modes. We used the Hilbert transform to extract the envelope of the interference signal to reduce the effect of the SNR waveform burr. Then EMD algorithm was used to separate the low and high-frequency components of the raw interference signal to reduce the influence of high-frequency components in spectrum estimation. There is a very good connection between the soil moisture retrieval based on the two models and the predetermined soil moisture. The multi-layer SMC retrieval model is more versatile than the conventional GNSS-IR single-layer SMC retrieval model in that it can assess the SMC at various soil depths. In order to understand the vertical distribution of soil water content and to direct the agricultural output, this study’s theoretical model serves as a guide. The lowest soil often has low measurement precision due to signal attenuation during soil propagation. Data indicate a correlation of 0.92 between the measured results and the specified soil moisture level.

The soil model used in this study is based on true SMC data using a simulation platform in order to calibrate the GNSS-IR multi-layer SMC measurement model proposed in this study and enhance the measurement’s accuracy. We will conduct experimental verification in subsequent work.

## Figures and Tables

**Figure 1 sensors-23-01949-f001:**
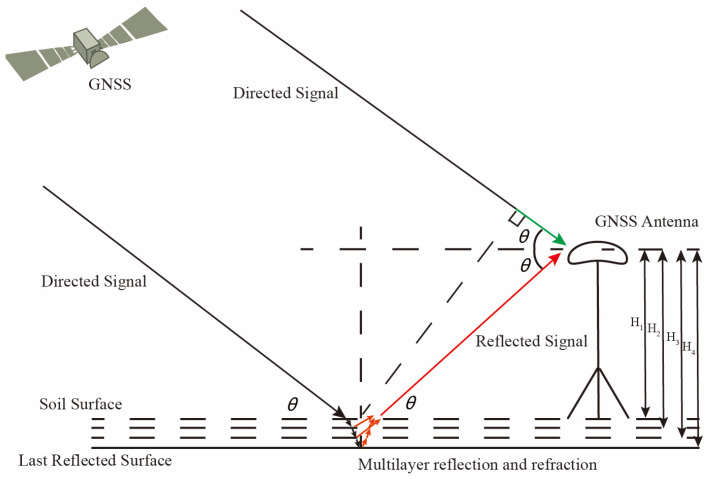
Schematic diagram of the propagation path of the GNSS signal in the multi-layer soil model.

**Figure 2 sensors-23-01949-f002:**
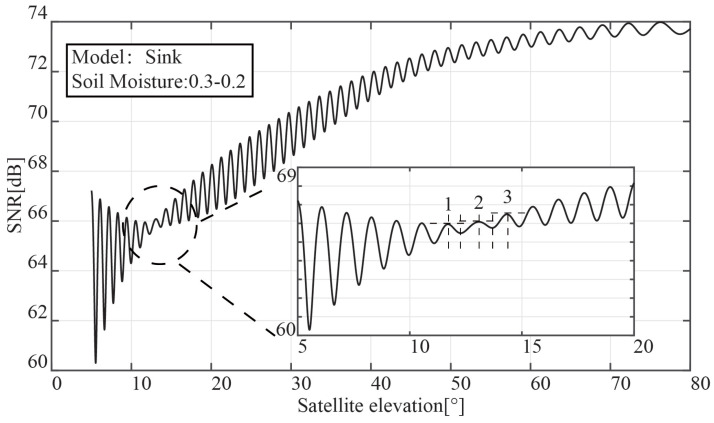
Vertically polarized interference signal–SNR oscillation waveform.

**Figure 3 sensors-23-01949-f003:**
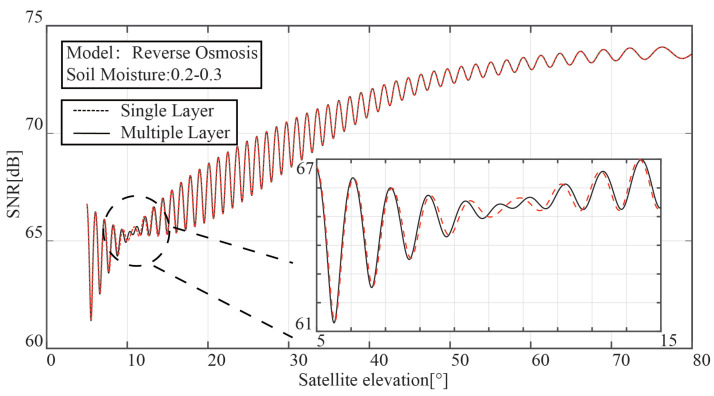
Interference signal comparison between the single-layer model and multi-layer model.

**Figure 4 sensors-23-01949-f004:**
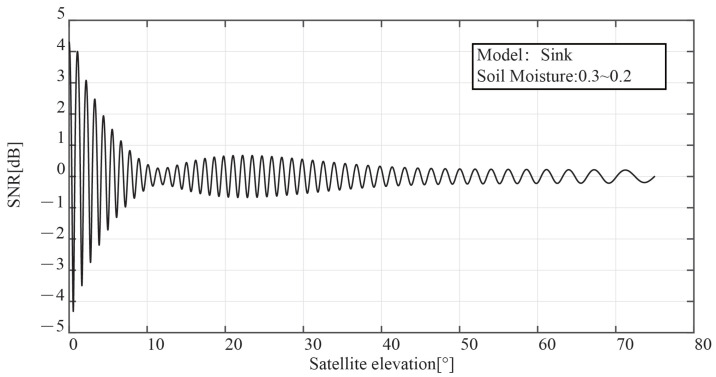
The SNR waveform of the reflected signal after decoupling the direct reflected signal.

**Figure 5 sensors-23-01949-f005:**
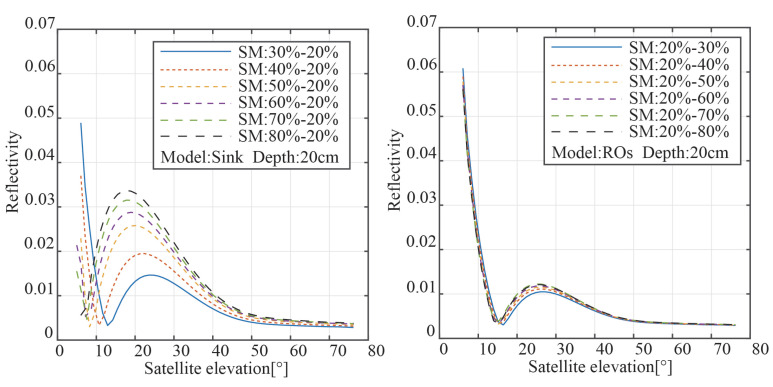
The power reflectivity results for the two modes with different moisture intervals.

**Figure 6 sensors-23-01949-f006:**
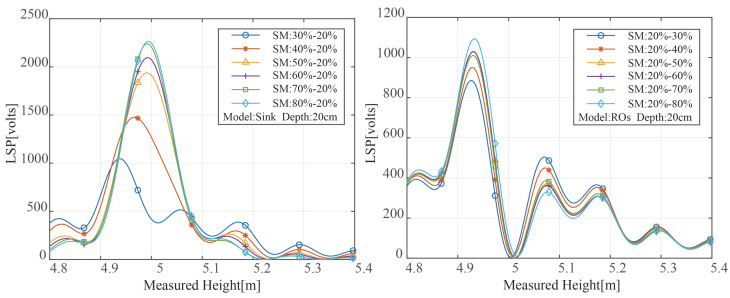
Spectrum of the two models with different soil moisture changes without noise.

**Figure 7 sensors-23-01949-f007:**
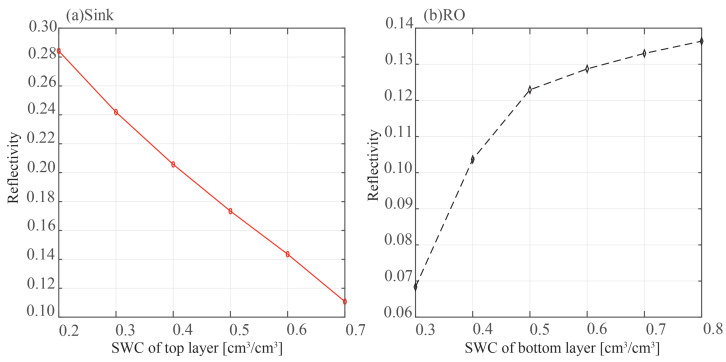
The reflectivity of the second layer changes with SMC.

**Figure 8 sensors-23-01949-f008:**
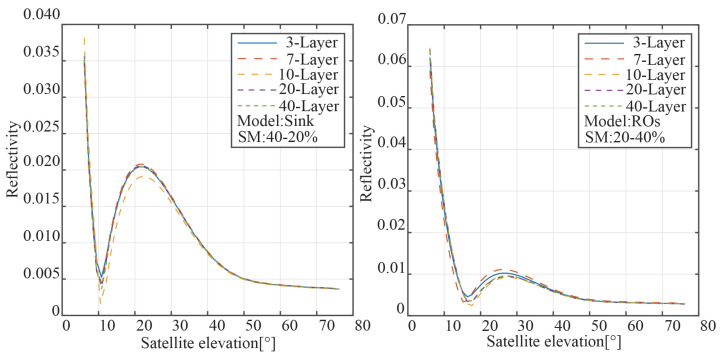
The power reflectivity results for the two modes with different layers.

**Figure 9 sensors-23-01949-f009:**
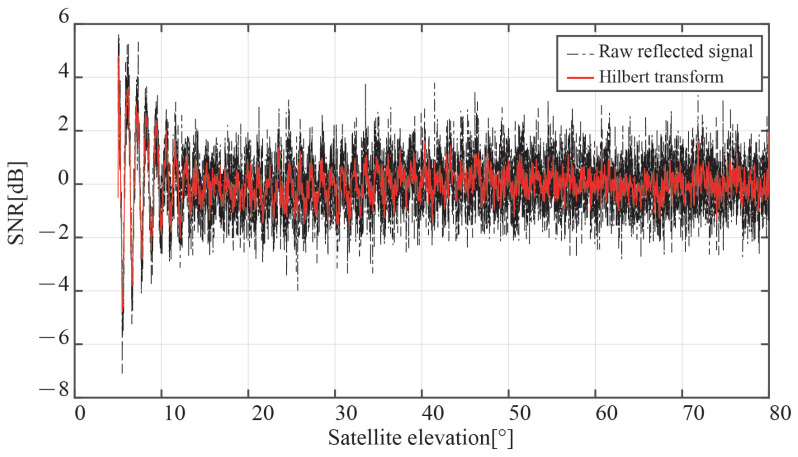
SNR time sequence of the interference signal with Gaussian white noise and its envelope by Hilbert transform.

**Figure 10 sensors-23-01949-f010:**
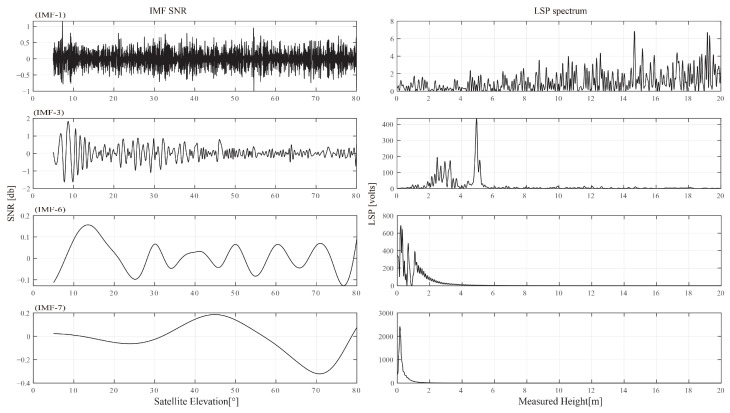
EMD decomposition results and spectral characteristics.

**Figure 11 sensors-23-01949-f011:**
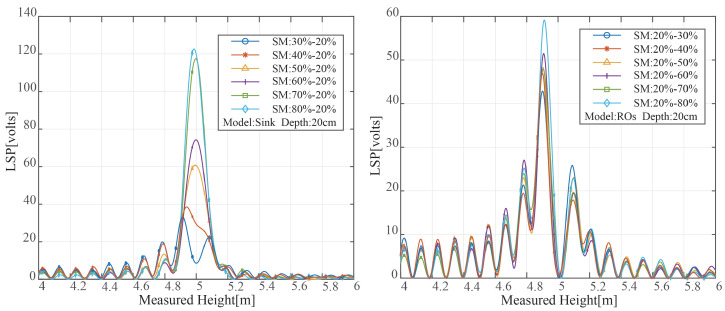
Spectrum of the two models with different soil moisture changes.

**Figure 12 sensors-23-01949-f012:**
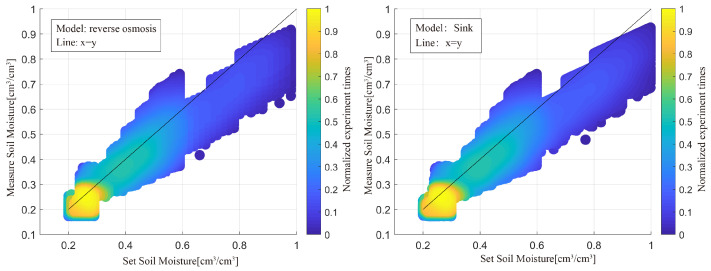
The scatter plot of the set SMC and measurement results for the two modes.

## Data Availability

Not applicable.

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
