# Peer review of "Multilayer Model in Soil Moisture Content Retrieval Using GNSS Interferometric Reflectometry"

_sensors, 2023, doi:10.3390/s23041949_

Round 1

Reviewer 1 Report

The submitted manuscript titled " Multilayer Model in Soil Moisture Content Retrieval Using GNSS Interferometric Reflectometry" is a valuable work. The subject is within the scope of the journal. Opinions, Methods, Results, Interpretation, Conclusions, Results The value and writing of the paper are reasonable. This research has reference value for the safety and management of the actual operation process of the project. Therefore, there is no objection to publishing this paper. There are several additional issues that need to be revised as follows:

(1) "Some researchers have looked at the retrieval of soil moisture under vegetation cover in an effort to lessen the impact of vegetation on soil moisture retrieval" suggests adding Some references to support the idea since you used" Some researchers"

(2) In the conclusion, there are less statements about the research results, while in the last paragraph of the introduction, there are more statements about the conclusions. It is suggested to rethink the content of these two parts.

(3) Please verify the correctness of formula (1) as it is not quite the same as the formula in reference (8).

(4) "When Figure 2 is included, it is clearthat the interference signal’s current "notch" position is between 10° and 20° of the satellite elevation angle" This sentence is a little difficult to understand, please consider revising it. The results from reverse osmosis in FIG. 5 are not described in this paper, and it is suggested to add relevant expressions.

(5) "Figure 3 shows that, unlike the sinking mode, the satellite elevation angle is bigger when the upper soil corresponds to entire transmission for the reverse osmosis method and is about 10° when full projection occurs" Whether this sentence appears near Figure 7 is an incorrect statement.

Reviewer 2 Report

Dear authors.

Introduction:

  • GNSS-IR has traditionally been used to monitor soil moisture content, assuming a single soil layer. However, since the soil moisture content is not evenly distributed along the longitudinal axis, the authors propose a stratified model applied at different depths. To do this, they use GNSS signals using linearly polarized electromagnetic waves and the Hilbert transform to analyze signal interference.
  • The authors extensively report the different models and technology to analyze the soil moisture content, considering the different characteristics of the soils. In addition, they develop the theory of signal interference by which they analyze the soil moisture content assumed to be made up of several layers at different depths.

  • Notwithstanding the foregoing, I must point out the following:
  • 1.- The article focuses mainly on technical-theoretical aspects of the proposed model
  • 2.- When reading the abstract, the reader assumes a priori that the research results are based on experiments carried out in the field. The authors must explicitly point out that the analysis carried out is through simulation, where they use the proposed theoretical model.
  • 3.- The content of the discussion and conclusion sections are quite similar. They should be merged into a single section and thus avoid unnecessary repetition of statements.

4.- Considering that the problem to be investigated is to assume multi-layer soil instead of just one layer, the difference in the results should be highlighted, pointing out the importance of the new proposed model compared to the one-layer model. Not just point to a correlation of 0.92; which at first might seem very poor.

  • Suggested improvements:

      Only some typing problems and punctuation marks I suggest checking:

Lines 102, 112 : Say:  Figure  ……………..   Change: Figure

Line 110: citep ¿?

After equation (2): Say Hrepresent ………… change: H represent

Line 119: Say:  Fennel                                   change: Fresnel.

After line 123 must say:    Fresnel.  And reflexionangle ……change: reflexion  angle.

Line 270-271:   ¿%cm3/cm3?

Figure 12: Indicate meaning of color scale.

Specify the meaning of SNR and SWC.

Round 2

Reviewer 2 Report

The authors have satisfactorily clarified all the questions and observations indicated.